# Cribriform versus Intraductal: How to Determine the Difference

**DOI:** 10.3390/cancers16112002

**Published:** 2024-05-24

**Authors:** Eva Compérat, Johannes Kläger, Nathalie Rioux-Leclercq, André Oszwald, Gabriel Wasinger

**Affiliations:** 1Department of Pathology, Medical University of Vienna, 1090 Vienna, Austria; 2Department Pathology, CHU de Rennes, 35033 Rennes, France

**Keywords:** prostate cancer, intraductal carcinoma, cribriform, pathological risk factors

## Abstract

**Simple Summary:**

Prostate cancer is a common and challenging disease among men, driving researchers to find better ways to understand and manage it. The understanding of two specific types, cribriform and intraductal prostate cancer (IDC-P), has evolved significantly over the years. This review aims to help pathologists differentiate between cribriform prostate cancer and IDC-P, and addresses current recommendations. Recent studies show that including these features in decision-making tools enhances predictions of cancer recurrence, spread, and patient outcomes. Future research should further focus on their pathological and molecular aspects to improve risk stratification, treatment approaches, and patient care.

**Abstract:**

Over the years, our understanding of cribriform and intraductal prostate cancer (PCa) has evolved significantly, leading to substantial changes in their classification and clinical management. This review discusses the histopathological disparities between intraductal and cribriform PCa from a diagnostic perspective, aiming to aid pathologists in achieving accurate diagnoses. Furthermore, it discusses the ongoing debate surrounding the different recommendations between ISUP and GUPS, which pose challenges for practicing pathologists and complicates consensus among them. Recent studies have shown promising results in integrating these pathological features into clinical decision-making tools, improving predictions of PCa recurrence, cancer spread, and mortality. Future research efforts should focus on further unraveling the biological backgrounds of these entities and their implications for clinical management to ultimately improve PCa patient outcomes.

## 1. Introduction

Cribriform and intraductal prostate cancer (PCa) have been known about for a long time, and their integration into classifications has evolved over time. Initially, Donald Gleason categorized all cribriform elements as Gleason pattern (GP) 3 in his early publications. However, through subsequent years and multiple consensus conferences, our understanding of these lesions and their significance have advanced [1].

From both a pathological and clinical perspective, it is crucial to distinguish between these two entities. Intraductal carcinoma of the prostate (IDC-P) is a non-invasive PCa. It is important to recognize whether there is an associated invasive component. Conversely, cribriform PCa (cPCa) is always an invasive carcinoma and is now classified as GP 4, which is considered aggressive. Based on their morphological similarities, IDC-P and cPCa are often treated as the same entities in the literature. This presents a challenge as it complicates the interpretation of research findings concerning either of these entities, potentially obscuring distinct clinical implications and treatment strategies. Furthermore, it is important to note that while one has a non-invasive feature, the other represents aggressive invasive prostate cancer, which may affect treatment decisions.

This review focuses on the differences between IDC-P and cPCa from a histopathological perspective, aiming to assist pathologists in reaching accurate diagnoses. 

## 2. Intraductal Carcinoma of the Prostate

The first description of IDC-P was carried out by Rhamy et al. in 1972 [2]. It was found again years later in a publication in 1985 by Kovi et al. who described these findings in transurethral resections of the prostate [3]. IDC-P was considered to represent an intraductal extension of invasive PCa at that time. We now have a better understanding of this entity, and in the latest version of the WHO (World Health Organization) classification of 2022, experts highlighted the duality of these lesions, meaning that most lesions are associated with invasive PCa, but some are pure intraductal forms [4]. However, pathologists agreed that pure IDC-P is a rare entity, constituting approximately 2% of IDC-P cases [5]. Since 2010, there have been reports describing IDC-P without any co-existing invasive component on needle biopsy, suggesting it might represent a stage of PCa associated with high-grade prostatic intraepithelial neoplasia (HGPIN) before the development of an invasive component [5,6]. In such cases, it is crucial to conduct a new series of biopsies in conjunction with mpMRI (multiparametric Magnetic Resonance Imaging) to rule out invasive PCa. The debate regarding surgery remains ongoing, with some authors advocating for radical prostatectomy (RPE) as a potential treatment [6]. 

From a histological standpoint, IDC-P is a complex lesion, characterized by an expansive proliferation of tumor cells within ducts and acini. Consequently, the basal layers of these structures remain intact, with no evidence of invasive growth into the prostatic stroma (see Figure 1A). Basal cells typically express markers such as p63 and HMWCK (high-molecular-weight cytokeratins) (see Figure 1B). Various architectural patterns have been documented, primarily cribriform growth, although solid or dense configurations have also been reported [7,8,9]. When referring to dense configurations, pathologists describe a growth pattern that appears more solid than cribriform, with sparse luminal spaces. It is crucial to note that IDC-P has consistently been linked with pleomorphic or hyperchromatic cells, exhibiting enlarged features. Mitosis and apoptosis have also been reported, although these aspects are more controversial in the literature [7,8,9]. Other morphological aspects that have been described include micropapillary features, although the most widely recognized features of IDC-P include solid or dense cribriform aspects, with a clear preservation of the basal layer, as seen in Figure 1.

One of the most surprising findings in recent literature is the presence of comedonecrosis in IDC-P, which seems to be particularly associated with this type of lesion [10,11]. This discovery naturally raises questions regarding numerous older pathology reports as well as reports in the literature. Prior to these findings, nobody suspected comedonecrosis to be associated with non-invasive tumoral proliferation. Consequently, one significant retrospective question that arises is the frequency with which comedonecrosis was diagnosed as GP 5 without evidence of invasiveness. 

It is now well known that patients with IDC-P should not be included in active surveillance (AS) programs and some national guidelines even officially implement this recommendation in their latest versions [12]. IDC-P in RPE displays a higher grade of an invasive component if present. Furthermore, the tumors seem to be larger and the probability of having a pT3 tumor with extraprostatic extension is higher. This is true for pT3a (periprostatic fatty tissue invasion) as well as pT3b (invasion of seminal vesicles), and pelvic lymph node metastases have been more frequently reported [13,14,15,16,17,18]. Furthermore, a systematic review of the literature conducted in 2017 by Porter et al. revealed an increase in IDC-P prevalence from 2.1% in low-risk PCa cohorts to 23.1%, 36.7%, and 56.0% in moderate-risk, high-risk, and metastatic or recurrent-disease risk categories, respectively [19]. It has also been shown that after RPE, in the case of a supplementary presence of IDC-P, the patient more frequently has a biochemical recurrence (BCR), a shorter progression-free survival, and a lower cancer-specific mortality [13,14,15,16,17,18]. Another adverse outcome is metastatic failure after radiation therapy in patients with intermediate and high-risk PCa [20,21,22,23].

On a molecular level, IDC-P often harbors genetic alterations, which are commonly associated with high-grade cancers, including the loss of PTEN or mutations in TP53 [24]. A significant elevation in the frequency of BRCA2 mutations is also reported [25,26] and some guidelines have recommended testing for BRCA2 mutations in the case of IDC-P, but this finding remains controversial [27,28]. These alterations contribute to the aggressive clinical characteristics of IDC-P. However, there seems to be a notable reduction in these molecular changes in cases of isolated IDC-P, suggesting a different origin or biological behavior of these entities compared to conventional IDC-P [29]. This highlights the importance of thorough molecular characterization for guiding clinical decisions and treatment strategies, as well as the need for further research on this topic, since original research articles on this are sparse [29].

## 3. Cribriform Prostate Cancer

Donald Gleason introduced the term “cribriform glands” to describe “glands composed of sheets of tumor cells that form cohesive rounded or irregularly shaped trabeculae with perforations or punched out lumina” (see Figure 2) [30]. The prevalence of cribriform morphology in PCa varies widely between studies, ranging from 9.3 to 37% in prostate biopsies [31,32], and from approximately 25 to 70% in RPE specimens [32,33,34,35]. These patterns vary in form, sometimes with irregular borders, but pathologists generally agree when identifying them, with recent studies enhancing consistency through detailed descriptions [4]. There has been an ongoing debate regarding small and large cribriform patterns, with some evidence existing that larger patterns may indicate a worse clinical prognosis, although clear cut-offs remain unclear [36].

It is important to note that in PCa the term “cribriform pattern” always accompanies a loss of basal cells and signifies invasive carcinoma. In 2012, Dong et al. [37] described ill-defined glands with poorly formed lumina and large cribriform glands with smooth borders, which were redefined as GP 4 after the consensus conference in 2005 [38], and tried to evaluate their prognosis. The author also showed that in RPE, cribriform patterns were an independent predictor of BCR and metastases. They read the slides of 1240 consecutive RPE specimens and could show that 34% of patients with classical Gleason score (GS) 6 (3 + 3) were upgraded to modified GS 7 or 8 by the ISUP criteria. These results validated the importance of recognizing cribriform glands, and when present, they predicted a poorer outcome. Further studies over the years confirmed these findings [39,40], and in 2016, all cribriform patterns, independent of their size, were included as GP 4 in the WHO classification [41]. This finding was voted for in 2014 during a consensus conference, which was held with pathologists, oncologists, urologists, and radiation oncologists [1].

Furthermore, studies indicate that including the presence of cPCa or IDC-P in prostate biopsy reports improves the predictive values of disease-specific survival and metastasis-free survival compared to using only the GS [42]. Therefore, it is crucial to mention these features in biopsy reports. Similar findings have been observed in RPE, suggesting that these unfavorable histologic findings may have greater clinical significance than simply defining the percentage of grade group (GG) 4 alone, particularly in patients who develop local or distant metastases [43]. Also, ductal adenocarcinomas with a cribriform pattern are more often associated with an advanced pathological stage than those without this feature after prostatectomy [44]. However, attempts to predict lymph node metastases based on histomorphological findings (IDC-P or cribriform), alongside Briganti and MSKCC nomograms, did not improve accuracy [45].

Some studies have attempted to identify cPCa using MRI [46]. While some have successfully demonstrated that these patterns can indeed be detected through imaging, we believe that detecting all lesions may pose a challenge. Therefore, we suggest that utilizing mpMRI followed by prostate biopsies may offer a more accurate approach. 

Similar to IDC-P, some authors also recently showed higher genomic instability in cPCa, but the literature is not exhaustive and only limited data are available [24,47,48,49]. This higher genomic instability likely disrupts key regulatory pathways in cPCas, affecting genes like *mTORC1*, *MYC*, *MAPK*, *KRAS*, and *JAK-STAT* [48]. Abnormal RNA expression, particularly overexpression of *SChLAP1*, is associated with increased metastatic potential [47]. Furthermore, patients with cPCa tend to fare worse on RNA-based tests like OncotypeDx Genomic Prostate Score^®^ (GPS) and Decipher [50,51]. Overall, these findings underline the importance of personalized risk assessments and tailored management approaches for patients with cPCa.

## 4. Differential Diagnoses

According to the literature, precursor lesions of PCa that might mimic IDC-P include HGPIN and atypical intraductal proliferations (AIP). Concerning HGPIN, the WHO classification of 2022 specifies that only high-grade lesions should be reported, as there has been significant discordance in the reporting of low-grade PIN [4].

HGPIN shows fewer atypical features than AIP and has long been viewed as the precursor to invasive PCa. Furthermore, HGPIN may also precede intra-acinar or intraductal spread, expanding its role as a precursor lesion. The WHO 2022 classification now includes cribriform patterns of HGPIN within the atypical intraductal proliferation group [4]. Molecularly, both HGPIN and invasive PCa show similar alterations such as TMPRSS2-ERG gene fusions, supporting this new consideration [52]. Some authors have described only partial ERG overexpression in HGPIN glands, suggesting that these positive cells may rapidly progress to invasive PCa. Additionally, the molecular relationship between HGPIN and adjacent PCa further underscores this theory [53]. The concept of HGPIN dates back to the 1980s when Bostwick and al. described it as “anaplasia of cells lining prostatic ducts and acini” with a conservation of the basal cell layer [54]. However, it is now believed that HGPIN may not be the direct precursor of invasive PCa, and instead AIP and IDC-P are considered intermediate steps before the development of invasive PCa. 

AIP represents a highly complex lesion, displaying greater cytological atypia than HGPIN but not as extensive as IDC-P. One of the main challenges is the uncertainty surrounding their clinical significance when detected on prostate biopsies. It is unclear whether patients should undergo additional biopsies or simply be monitored closely [55]. 

Some precursor lesions can have overlapping features such as cribriform HGPIN, and it can be extremely difficult to determine the difference between IDC-P and cPCa, but consensus papers have helped clarify these distinctions [56,57]. A recent publication regarding prostate needle biopsies established clear diagnostic criteria for identifying cribriform patterns [56]. The authors aimed to establish a large set of consensus cases for standardization. Briefly, the consensus criteria were as follows: more than two cribriform structures per level, or the largest cribriform mass with ≥9 lumina, or a diameter of ≥0.5 mm. These aspects predicted a consensus diagnosis of cribriform cancer in over 80% of cases. Given the importance of the outcome, standardization is of major importance. 

Additionally, it is crucial to recognize that cribriform aspects may occasionally appear in the center of the prostate without significant cytological atypia [58]. Recognizing the difference between this and urothelial carcinoma invading the prostate can be challenging, as can differentiating it from solid IDC-P spreading into the ducts and acini.

Another challenging entity in the differential diagnosis of IDC-P is ductal PCa, which is a separate chapter in the WHO 2022 classification [4]. It is also characterized by intraductal growth, but normally the ductal component is adjacent and invasive. Essential and desirable diagnostic criteria are the following: identification of glandular structures with papillary and/or complex cribriform morphology lined by tall columnar pseudostratified cells; often (but not always) high-grade nuclear atypia; in RPE: >50% histology (and the percentage reported); in a needle biopsy: even if pure, use the terminology “adenocarcinoma of prostate with ductal features” [4].

In case of any doubt as to whether the tumor is invasive cPCa or still IDC-P, immunohistochemistry (IHC) is of help. Most of the time, a double stain with p63 (a nuclear marker of the basal layers) and AMACR (alpha-methylacyl-CoA racemase, a cytoplasmatic marker of tumor cells) will be co-expressed in IDC-P (see Figure 1) and, therefore, together with morphology, guide the pathologist toward the right diagnosis [4]. For an overview of the different IHC expressions in cPCa, IDC-P, and mimickers, see Table 1.

A current ongoing controversy is whether IHC should be used in prostate biopsies if an invasive component is present and a suspicion of IDC-P exists. Some clinicians think that this finding impacts the assigned prostate grade, and there is still an ongoing debate in pathology as to whether the presence of IDC-P should be included in the GS. The two existing uropathology societies do not recommend the same reporting of these specimens [1,59]. Most of the time, IHC is not necessary, as in the case of an invasive carcinoma, where the supplementary presence of IDC-P will not change the treatment strategy, but a global recommendation is to report IDC-P in prostate biopsies as well as in RPE, as an adverse outcome is linked to it. As already mentioned above, patients with IDC-P should not be included in AS programs, as this finding is linked to adverse pathological findings [12]. Nevertheless, staging of PCa is only possible if an invasive component exists on an RPE.

## 5. Current Recommendations for IDC-P and cPCa

The 2019 grading changes proposed by ISUP and GUPS are yet to be fully validated, and there are also some specific differences between the recommendations from the two bodies, which current evidence cannot yet resolve [1,59,60]. Both organizations advocate reporting the presence of invasive cPCa in GS 7 and 8 cases (GG 2–4) due to its prognostic significance [31,35,61,62,63,64]. However, precisely defining invasive cPCa poses challenges in reproducibility, particularly in distinguishing between small and large cribriform glands, as well as consistent distinction from IDC-P without using IHC [35,36,65,66]. Excluding IDC-P from Gleason grading may also be problematic without more widespread use of IHC in routine practice [10,11,67]. While awaiting further definitive evidence to reconcile these differences between the 2019 ISUP and GUPS proposals, pathologists should specify which version of the Gleason grading system recommendations they are using in reports and publications to allow meaningful analyses and comparisons of cohorts.

A unified understanding and identification of these entities are especially crucial in light of a recent publication attempting to integrate these pathological features into existing clinical decision-making tools. These efforts have shown promising results in predicting PCa recurrence, cancer spread, and death due to disease in a multicenter cohort of 1326 patients [68]. Despite some limitations of the study such as its retrospective nature and inclusion of cases only reported by genitourinary pathologists, the results are promising and underscore the relevance of these entities for patient stratification.

## 6. Conclusions and Outlook

In recent years, our understanding of cPCa and IDC-P has advanced significantly, leading to notable changes in their classification and clinical management. Initially categorized as GP 3, recognition of the prognostic significance of cribriform lesions has led to their re-evaluation and being classified as GP 4 since the WHO 2016 classification. Intraductal carcinoma of the prostate (IDC-P) represents a non-invasive PCa entity, often associated with invasive components. Despite being a rare entity, pure IDC-P has also been identified, prompting the need for thorough diagnosis through additional biopsies and imaging modalities. The ongoing controversy surrounding the differences of the current recommendations between ISUP and GUPS poses challenges for practicing pathologists and complicates consensus among them. While awaiting further definitive evidence to reconcile these differences between the 2019 ISUP and GUPS proposals, pathologists should specify which version of the Gleason grading system recommendations they are following.

In conclusion, the evolving landscape of cPCa and IDC-P underscores the importance of comprehensive histopathological evaluation, molecular characterization, and multimodal diagnostic approaches. Recent studies have shown promising results in integrating these pathological features into clinical decision-making tools, improving predictions of PCa recurrence, cancer spread, and mortality. Future research efforts should focus on further unraveling the biological backgrounds of these entities and their implications for clinical management to ultimately improve PCa patient outcomes.

## Figures and Tables

**Figure 1 cancers-16-02002-f001:**
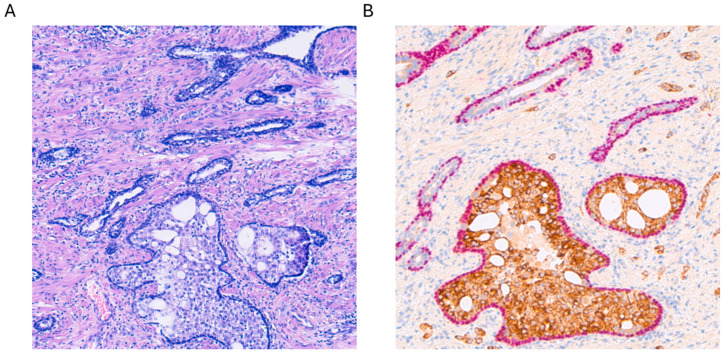
(**A**) Hematoxylin and eosin staining of intraductal carcinoma of the prostate (IDC-P) at 100× magnification. (**B**) Alpha-methylacyl-CoA racemase (AMACR, brown) and p63 (pink) double staining with hemotoxylin counterstaining of the same lesion with strong positive cytoplasmatic AMACR staining accompanied by nuclear expression of p63 in the basal cell layer, which is typical for IDC-P.

**Figure 2 cancers-16-02002-f002:**
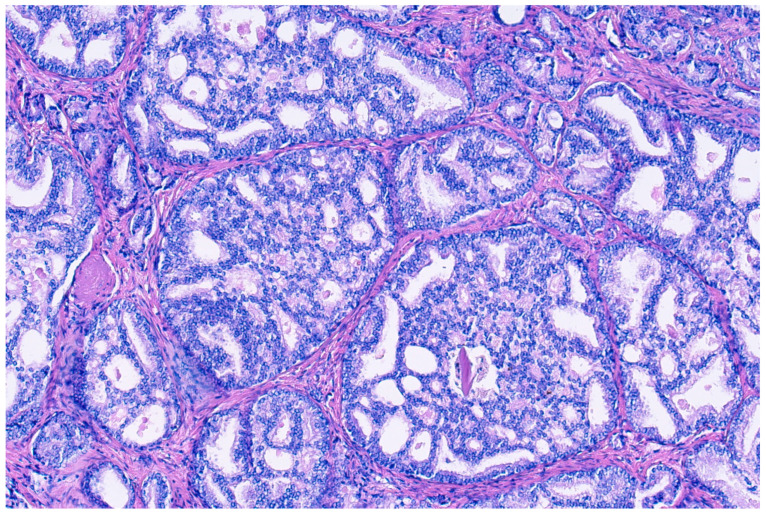
Hemotoxylin and eosin staining at 100× magnification of cribriform prostate cancer glands composed of sheets of tumor cells that form cohesive rounded or irregularly shaped trabeculae with perforations or punched out lumina.

**Table 1 cancers-16-02002-t001:** Overview of p63 and AMACR (alpha-methylacyl-CoA racemase) expression in HG-PIN (high-grade prostatic intraepithelial neoplasia), AIP (atypical intraductal proliferation), IDC-P (intraductal carcinoma of the prostate), cPCa (cribriform prostate cancer), and acinar PCa.

Lesion Type	p63	AMACR
HG-PIN	+	+
AIP	+	+ weak/on surface
IDC-P	+	+
cPCa	−	+
Acinar PCa	−	+

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
