# Peer review of "Cribriform versus Intraductal: How to Determine the Difference"

_cancers, 2024, doi:10.3390/cancers16112002_

Round 1

Reviewer 1 Report

Comments and Suggestions for Authors

The authors should be congratulated for their work. The review is organized as a practical guideline to assist pathologists in the diagnosis of molecular prostate cancer (PCA), specifically intraductal (IDC) vs. classic adenocarcinoma. The work is comprehensively edited and well-structured. It sounds interesting and worthy of further discussion of the differencial diagnosis between the two entities (precursor lesions of PCa, that might mimic IDC-P, such as HGPIN and atypical intraductal proliferations). Specifically, to provide instruments also to the practical management of these conditions, it would be interesting to add information regarding:

- the prevalence of both entities in incidental PCA and technique to enhance their diagnosis (in order to provide direct management advice for the urologists/radiotherapists who need to  perform surgery or RT);

- future perspective to better contextualize how the research is evolving to define a prompt diagnosis for these patients, who have different survival outcomes, in order to ensure the best time for treatment.

    Author Response

    Thank you for your positive feedback and insightful comments on our work titled “Cribriform versus Intraductal, How to Make the Difference.". We appreciate your suggestions and have made updates to our manuscript to address them:

    We added detailed information regarding the prevalence of intraductal carcinoma of the prostate (IDC-P) and cribriform morphology to the respective chapters. These additions aim to enhance the practical utility of our guidelines for pathologists and clinicians by providing a clearer picture of how often these features are encountered in various diagnostic contexts.

    While we recognize the importance of your comment about providing direct management advice for urologists and radiotherapists, our experience indicates that the choice between radiotherapy and surgery is highly dependent on the overall staging of the disease rather than the mere presence of IDC-P or cribriform morphology. Furthermore, management decisions are significantly influenced by patient-specific factors and the preferences of the practicing clinician. Thus, while we can outline general principles, individualized treatment planning remains paramount.

    Currently, the field is not witnessing any significant advancements that would allow us to provide additional information on evolving research. However, we believe it is crucial to differentiate between IDC-P with and without an invasive component, as well as cribriform invasive prostate carcinoma. This distinction could potentially influence future research directions and clinical management strategies.

    Additionally, I would like to mention, that we just recently published a paper specifically addressing the clinical management of IDC-P. This publication provides a comprehensive overview and could serve as a valuable resource for clinicians. Here is the link to the paper: doi: 10.3390/cancers16091650

    We hope these updates and clarifications meet your expectations and enhance the utility of our review for its intended audience. Thank you once again for your valuable input.

    Sincerely,

    Gabriel Wasinger

    Reviewer 2 Report

    Comments and Suggestions for Authors

    I have read the manuscript titled "Cribriform versus intraductal, how to make the difference." This paper discusses the histopathological characteristics distinguishing intraductal from cribriform prostate cancer. In general, I found this paper to be well-written. It provides up-to-date knowledge regarding prostate cancer diagnosis from a histopathological point of view. I believe both pathologists and urologists could benefit from this review paper, and the authors should therefore be complimented for their excellent work. In my opinion, no revisions are required, and the manuscript should be accepted in its current form.

    Author Response

    Thank you very much for your positive feedback on our manuscript titled “Cribriform versus Intraductal, How to make the Difference.”. We are delighted to hear that you found the paper well-written and informative. Your recommendation for acceptance without revisions is highly encouraging. We hope our work will indeed be valuable to both pathologists and urologists. Thank you once again for your kind words and support.

    Sincerely,

    Gabriel Wasinger